# Low Doses of Silver Nanoparticles Selectively Induce Lipid Peroxidation and Proteotoxic Stress in Mesenchymal Subtypes of Triple-Negative Breast Cancer

**DOI:** 10.3390/cancers13164217

**Published:** 2021-08-22

**Authors:** Christina M. Snyder, Monica M. Rohde, Cale D. Fahrenholtz, Jessica Swanner, John Sloop, George L. Donati, Cristina M. Furdui, Ravi Singh

**Affiliations:** 1Department of Cancer Biology, Wake Forest School of Medicine, Winston Salem, NC 27157, USA; cmsnyder@wakehealth.edu (C.M.S.); mmcmahon@wakehealth.edu (M.M.R.); cale@highpoint.edu (C.D.F.); jswanner16@gmail.com (J.S.); 2Department of Chemistry, Wake Forest University, Winston-Salem, NC 27109, USA; sloojt16@wfu.edu (J.S.); donatidl@wfu.edu (G.L.D.); 3Department of Internal Medicine, Section of Molecular Medicine, Wake Forest School of Medicine, Winston-Salem, NC 27157, USA; cfurdui@wakehealth.edu; 4Comprehensive Cancer Center of Wake Forest Baptist Medical Center, Winston Salem, NC 27157, USA

**Keywords:** silver nanoparticle, triple-negative breast cancer, proteotoxic stress, lipid peroxidation, precision medicine, nanomedicine, claudin low, EMT

## Abstract

**Simple Summary:**

Different types of breast cancer are typically classified based upon expression of estrogen receptor, progesterone receptor, and epidermal growth factor receptor. One aggressive form of breast cancer is called triple-negative breast cancer (TNBC) because it lacks expression of these proteins. However, TNBC is also heterogeneous and can be further divided into distinct classes including an epithelial-like group and a mesenchymal-like group. The mesenchymal subtype may be vulnerable to therapeutic strategies that oxidize lipids and proteins. The aim of our study was to determine if silver nanoparticles (AgNPs) can cause these types of damage to selectively kill mesenchymal TNBCs. We found that AgNPs killed mesenchymal TNBCs by a mechanism involving lipid and protein oxidation without causing similar toxicity to normal breast cells. This study shows AgNPs are a specific treatment for mesenchymal TNBCs and indicates that stratification of TNBC subtypes may lead to improved outcomes for other therapeutics with similar mechanisms of action.

**Abstract:**

Molecular profiling of tumors shows that triple-negative breast cancer (TNBC) can be stratified into mesenchymal (claudin-low breast cancer; CLBC) and epithelial subtypes (basal-like breast cancer; BLBC). Subtypes differ in underlying genetics and in response to therapeutics. Several reports indicate that therapeutic strategies that induce lipid peroxidation or proteotoxicity may be particularly effective for various cancers with a mesenchymal phenotype such as CLBC, for which no specific treatment regimens exist and outcomes are poor. We hypothesized that silver nanoparticles (AgNPs) can induce proteotoxic stress and cause lipid peroxidation to a greater extent in CLBC than in BLBC. We found that AgNPs were lethal to CLBCs at doses that had little effect on BLBCs and were non-toxic to normal breast epithelial cells. Analysis of mRNA profiles indicated that sensitivity to AgNPs correlated with expression of multiple CLBC-associated genes. There was no correlation between sensitivity to AgNPs and sensitivity to silver cations, uptake of AgNPs, or proliferation rate, indicating that there are other molecular factors driving sensitivity to AgNPs. Mechanistically, we found that the differences in sensitivity of CLBC and BLBC cells to AgNPs were driven by peroxidation of lipids, protein oxidation and aggregation, and subsequent proteotoxic stress and apoptotic signaling, which were induced in AgNP-treated CLBC cells, but not in BLBC cells. This study shows AgNPs are a specific treatment for CLBC and indicates that stratification of TNBC subtypes may lead to improved outcomes for other therapeutics with similar mechanisms of action.

## 1. Introduction 

Breast cancer is a heterogeneous disease consisting of multiple subtypes. Histopathological tumor analysis identifies 15–20% of breast cancers cases as triple-negative breast cancer (TNBC), characterized by lack of expression of estrogen receptor, progesterone receptor, and low expression of human epidermal growth factor receptor (HER2) [1]. Based upon differences in global mRNA expression, TNBC subdivides into distinct, intrinsic molecular subtypes, defined by Perou and colleagues, as claudin-low breast cancer (CLBC) and basal-like breast cancer (BLBC) [2]. These subtypes largely overlap with Lehmann’s classifications [3] in which CLBC exhibits features of the mesenchymal (M) subtype and BLBC exhibits features of basal-like 1 (BL1) and basal-like 2 (BL2) subtypes. BLBC represents the majority of TNBCs, but approximately one-third of TNBCs and 14% of all invasive breast tumors are CLBC, a poor prognosis, highly invasive, metastatic, and treatment-resistant subtype that is frequently associated with metaplastic and medullary clinical phenotypes [4,5,6]. CLBC/M and BLBC/BL1/BL2 not only differ on the molecular level, but also in incidence, treatment response, and patient survival [2,3,7,8,9]. Despite this heterogeneity, TNBC is treated as a single disease, with standard of care consisting of a combination of chemotherapy, radiation, and surgery. 

BLBC/BL1/BL2 cells exhibit epithelial features, and CLBC/M cells are enriched for features indicative of epithelial-mesenchymal transition (EMT), a conserved and dynamic process involved in embryonic development, wound healing, and cancer development and progression [2,3,6,8]. The EMT program causes a loss of epithelial characteristics, including apical-basal polarity and tight junction proteins expression (e.g., E-cadherin, claudins, occludins) and increased expression of mesenchymal transcription factors and other markers (e.g., zinc finer e-box binding protein (ZEB)1/2, Twist, Snail, Slug, N-cadherin, vimentin, fibronectin, collagen). Changes associated with EMT contribute to increased invasive capacity, metastatic spread, and resistance to chemo-, immuno-, and targeted therapy [10]. 

Although the EMT phenotype is associated with drug resistance [11], it may also expose targetable vulnerabilities. For example, TNBC cells that have undergone EMT increase synthesis and secretion of extracellular matrix (ECM) proteins, and the high baseline endoplasmic reticulum stress caused by the burden of ECM production renders them sensitive to agents that increase protein misfolding and induce proteotoxicity [12,13]. Additionally, EMT in head and neck [14], lung [15], and breast cancers [15,16,17] is correlated with a distinct lipid metabolic signature that results in enrichment of cell membranes with long chain, poly-unsaturated fatty acids (PUFAs). This characteristic makes these cells vulnerable to agents that induce lipid peroxidation [18]. Therefore, therapeutic strategies that increase lipid peroxidation and cause accumulation of misfolded or aggregated proteins may be particularly effective for mesenchymal TNBC subtypes such as CLBC. 

Use of nanotechnology is a promising approach to develop more specific, less toxic cancer therapies. An emerging concept in cancer nanotechnology is harnessing the intrinsic cytotoxicity of nanoparticles to modulate the dynamics of EMT [19]. Drug-free, inorganic nanomaterials composed of gold [20], gadolinium [21], or silver [22] may specifically target the EMT cancer state. We and others found that certain breast cancer cell lines are more sensitive to silver nanoparticle (AgNP) exposure than non-malignant mammary epithelial cells [23,24,25,26]. Notably, AgNP exposure increases protein oxidation and proteotoxic stress [23,24,25,27,28,29] and may cause lipid peroxidation in certain cells and organisms [30,31,32]. However, there is considerable heterogeneity in the relative sensitivity of various cancers to AgNP exposure [23,24,25,27,29,33,34], and it is not known if cancers with an EMT phenotype are more susceptible than other cancers or normal cells to proteotoxic stress or lipid peroxidation caused by exposure to AgNPs.

Here, we aim to determine if mesenchymal and epithelial subtypes of TNBC differ in sensitivity to AgNPs, and to define the vulnerabilities driving this difference. Therefore, we investigated effects of AgNPs on viability, proliferation, and clonogenic growth using a panel of TNBC cells representing CLBC and BLBC subtypes and non-malignant breast epithelial cells. We determined the correlation between the AgNP IC_50_ and mass of AgNP taken up, sensitivity to silver cations (Ag^+^), proliferation rate, and mRNA profile for multiple cell lines. We quantified lipid peroxide generation, formation of 4-hydroxynonenol (4-HNE) protein adducts, protein thiol oxidation, protein aggregation, and activation of proteotoxic stress signaling responses after AgNP exposure to identify differences in the mechanism of action of AgNPs in CLBC and BLBC. These studies will help to establish a rationale for the use of AgNPs as a therapeutic for CLBCs and guide future studies pertaining to the mechanism of action of AgNPs. 

## 2. Materials and Methods 

### 2.1. Silver Nanoparticles 

Spherical AgNPs, 25 nm in diameter, and stabilized with polyvinylpyrrolidone (PVP) were purchased as dried powders from nanoComposix, Inc (San Diego, CA, USA). Nanoparticles were dispersed by bath sonication in phosphate buffered saline (PBS), pH 7.4, without calcium or magnesium (Invitrogen, Carlsbad, CA, USA), at a concentration of 5 mg/mL based upon the mass of silver contained in the nanoparticles, and were then diluted in cell culture medium to the final concentration listed in the figures prior to addition to wells containing cells. The physicochemical properties including hydrodynamic diameter, colloidal stability in cell culture media, and ζ-potential of this material were characterized previously [24].

### 2.2. Cell Culture 

BT-20, HCC70, MDA-MB-468, BT-549, MDA-MB-231, MDA-MB-436, and MCF-10A cells were purchased from the ATCC (Manassas, VA, USA). SUM159 cells were purchased from Astrerand (now BioIVT, Westbury, NY, USA) and iMEC cells were provided by Dr. Elizabeth Alli in the Department of Cancer Biology at Wake Forest School of Medicine. Cell lines were expanded, and low passage stocks were stored in liquid nitrogen and maintained by the Wake Forest Comprehensive Cancer Center Cell Engineering Shared Resource. Cell lines and growth media are listed in Table 1.

All cells were verified to be free from mycoplasma contamination by routine testing using the MycoAlert Mycoplasma Detection Kit (Lonza, Morristown, NJ, USA). Cells were passaged and medium was changed twice weekly. Cell monolayers were grown on tissue culture treated plastics purchased from Corning Life Sciences (Corning, NY, USA) or on glass coverslips (Warner Instruments Corporation, Hamden, CT, USA). For live fluorescence imaging studies, cells were grown in 8-well chamber slides (EMD Millipore, Burlington, MA, USA). Cells were maintained in culture for no longer than 4 months before new cultures were established from low-passage frozen stocks. 

### 2.3. MTT Assay 

Cells were seeded on 96-well plates at a density of 3500–5000 cells per well (depending upon cell line) in 100 µL of complete media, recovered for 24 h, and then were exposed to AgNPs or AgNO_3_ (Sigma-Aldrich, St. Louis, MO, USA) in 100 µL of complete media containing doses of AgNPs or AgNO_3_ as listed in the figures. After 72 h, media containing AgNPs or AgNO_3_ were replaced with 200 µL of media containing 0.5 mg/mL 3-(4,5- dimethylthiazol-2-yl)-2,5-diphenyltetrazolium bromide (MTT; Sigma-Aldrich) and incubated for 1 h at 37 °C. Medium was removed by vacuum suction, and cells were lysed in 200 µL of DMSO and absorbance read using a Molecular Devices Emax Precision Microplate Reader (San Jose, CA, USA) at 560 nm. Absorbance measurements at 650 nm were subtracted from each reading to correct for any inconsistencies in optical properties between wells.

### 2.4. Clonogenic Growth Assay 

Cells were seeded on 6-well plates at a density of 300–1000 cells per well (depending upon cell line) in 4 mL of complete media, recovered for 24 h, and were then exposed to various concentrations of AgNPs and diluted in 500 uL of complete media, resulting in the dose listed in the figures. After 24 h, AgNP-containing media was replaced with fresh media and colonies were allowed to form for 14–21 days. During this time, culture media was replaced every 2–3 days. Plates were then washed with PBS, fixed for 10 min with 1:1:8 (vol:vol:vol) glacial acetic acid:methanol:water, stained for 10 min with crystal violet, and then washed in tap water. After drying, colonies of at least 50 cells were counted manually.

### 2.5. IncuCyte Proliferation Assay 

Cells were seeded on 96-well plates at a density of 3500–5000 cells per well (depending upon cell line) in 100 µL of complete media, recovered for 24 h, and then exposed to AgNPs in 100 µL of complete media containing doses of each drug listed in the figures. Plates were then placed in an IncuCyte ZOOM (Essen Bioscience, Ann Arbor, MI, USA) incubator and cell proliferation and morphology was monitored by time-lapse imaging for 72 h. The logarithmic growth phase of vehicle (PBS)-treated proliferation curves for each cell line was identified and doubling time was determined experimentally.

### 2.6. mRNA Expression 

RNASeq expression data for specific mRNAs were obtained from the publicly available Broad-Novartis Cancer Cell Line Encyclopedia (CCLE) which can be accessed here: https://portals.broadinstitute.org/ccle. The data set was uploaded on 29 September 2018 and accessed by us on 2 January 2019. 

### 2.7. Inductively Coupled Plasma Mass Spectrometry (ICP-MS) 

Cells were seeded in 6 cm dishes at a density of 500,000 cells in 4 mL of media. Cells were treated with AgNPs or PBS for 24 h and were then trypsinized, washed twice in PBS, pelleted, and stored at −20 °C. Samples were then digested with 1 mL of concentrated HNO_3_, 2 mL of 30% *v*/*v* H_2_O_2_, and 7 mL distilled-deionized water using a microwave-assisted digestion system (Ethos UP, Milestone, Sorisole, Italy). The digested samples were diluted to a final acid concentration of 2% *v*/*v* before Ag determination by ICP-MS. Trace metal grade HNO_3_ (Fisher, Pittsburgh, PA, USA), low trace metals H_2_O_2_ (Veritas, Columbus, OH, USA), and type 1 water (18 MΩ cm, Milli-Q^®^, Millipore) were used to digest samples and prepare all solutions. Reference solutions of silver were prepared in 2% *v*/*v* acid (HNO_3_) from a 1000 mg/L Ag stock (SPEX CertPrep, Metuchen, NJ, USA) and used for calibration. Analyses were performed using a tandem ICP-MS (8800 ICP-MS/MS, Agilent) equipped with a SPS 4 automatic sampler, a Scott-type double pass spray chamber operated at 2 °C, and a Micromist concentric nebulizer. The ^109^Ag isotope was monitored in single quadrupole mode. To minimize potential spectral interferences, helium gas (≥99.999% purity, Airgas) was used in the ICP-MS’s collision/reaction cell during data acquisition. Instrument operating conditions were: 1550 W radio frequency applied power, 10.0 mm sampling depth, 1.05 L/min carrier gas flow rate, and 4.0 mL/min collision/reaction gas flow rate.

### 2.8. Lipid Peroxidation Assays 

Cells were seeded on 8-well coverslip-bottom chamber slides at a density of 30,000 cells per well in 400 μL of complete medium for microscopy experiments, or on 6-well plates at a density of 500,000 cells in 3 mL of complete medium for flow cytometry experiments. Cells were allowed to adhere for 48 h and were then exposed to AgNPs for 24 h at 37 °C. For microscopy experiments, medium was removed and 10 µM Liperfluo dye (Dojindo Molecular Technologies, Rockville, MD, USA) diluted in 500 µL live cell imaging solution (Molecular Probes, Eugene, OR, USA) was added to each well. After 30 min Liperfluo-containing media was replaced with fresh live cell imaging solution and re-placed in incubator for 4 h. Fluorescence was then observed using an Olympus FV1200 spectral laser scanning confocal microscope. Fluorescence was quantified using ImageJ software. For flow cytometry experiments, medium was removed and 10 µM Liperfluo dye (Dojindo Molecular Technologies) diluted in 1 mL PBS was added to each well for 30 min. Cells were then washed with PBS, trypsinized, trypsin neutralized with complete media, and resuspended in fresh PBS. Cell fluorescence was measured within an hour of staining using a BD FACS Canto II Analyzer. Data analysis was performed using FCS Express 7 software (De Novo Software, Pasadena, CA, USA).

### 2.9. Western Blots 

Cells were plated on 6 cm dishes at a density of 1,000,000 cells in 4 mL of complete medium. Cells were allowed to recover for 48 h and then were exposed to AgNPs for 3 or 24 h at 37 °C. Medium was removed and cells were washed twice with ice cold PBS before lysis using Mammalian Protein Extraction Reagent supplemented with Halt Protease & Phosphatase Inhibitor Cocktail (both from Thermo Fisher Scientific). Protein concentration was determined for each sample using a Pierce bicinchoninic acid (BCA) protein assay kit (Thermo Fisher Scientific) according to manufacturer’s instructions. Proteins were size fractionated by gel electrophoresis and then transferred to a nitrocellulose membrane (Thermo Fisher Scientific). Non-specific binding was blocked by incubation for 30 min at room temperature with tris-buffered saline containing 0.1% Tween-20 (Bio-Rad) and either 5% blotting grade blocker (Bio-Rad) or 5% bovine serum albumin (BSA; Sigma-Aldrich). Membranes were incubated overnight at 4 °C in dilutions containing TBS-T and either 5% blotting grade blocker or 5% BSA and 1:1000 primary antibody. Primary antibodies used include: phospho-eIF2α (9721), eIF2α (9722), CHOP (2895), phospho-JNK (9255), JNK (9252), caspase-3 (9662), caspase-7 (9492), caspase-9 (9502), GAPDH (2118), β-actin (4970) purchased from Cell Signaling Technologies (Danvers, MA, USA), and 4-HNE (MA5-27570) purchased from Invitrogen (Rockford, IL, USA). Membranes were washed and then incubated with 1:1000 anti-rabbit (Cell Signaling Technologies) or anti-mouse (Cell Signaling Technologies) horseradish peroxidase (HRP)-conjugated secondary antibodies diluted in 5% blotting grade blocker in TBS-T for 1 h at room temperature. Immunoreactive products were visualized by chemiluminescence using SuperSignal Femto West reagent (Thermo Fisher Scientific). 

### 2.10. Protein Aggregation Assays 

Cells were grown on 18 mm coverslips in 12-well plates at a density of 250,000 cells in 1 mL of complete medium. Cells were allowed to recover for 48 h and then were exposed to AgNPs for 24 h at 37 °C. Medium was removed and cells were fixed with 4% formaldehyde solution and permeabilized (0.5% Triton X-100, 3 mm EDTA, pH 8.0). Cells were then stained for 30 min with Proteostat Aggresome Detection Reagent (1:1000) and Hoechst 33342 (1:500) (Enzo BioSciences, Ann Arbor, MI, USA) diluted in PBS, and then washed twice with PBS, and coverslips were mounted on glass slides with ProLong Gold antifade reagent (Invitrogen). Fluorescence was visualized using an Olympus FV1200 spectral laser scanning confocal microscope. Fluorescence was quantified using ImageJ software. 

### 2.11. Protein Oxidation Assays 

Cells were plated at a density of 500,000 cells per well on 6-well plates in 2 mL of complete medium and allowed to adhere for 24 h. Cells were exposed to AgNPs for 24 h. Afterward, fresh media containing 50 µM of DCP-NEt_2_-Coumarin (DCP-NEt_2_C) was added for 30 minutes. Cells were then washed twice with PBS, fixed with 100% methanol and fluorescence was measured using a FACS Canto II Analyzer (BD Biosciences). Analysis of the data was performed using FCS express version 7 (De Novo Software, Glendale, CA, USA).

### 2.12. ImageJ Analysis 

Corrected total cell fluorescence (CTFC) was calculated from grayscale images: (1)CTFC=Integrated density−(Area of selected cell×Mean pixel intensity of background)

### 2.13. Statistical Analysis 

Analysis was performed as described in the figure legends using Prism 9.1.1 software (GraphPad, San Diego, CA, USA). Number of technical and biological replicates used for each experiment is included in the figure legends.

## 3. Results

### 3.1. AgNPs Are Lethal to CLBCs at Doses That Have Little Effect on BLBCs and No Effect on Viability of Non-Malignant Breast Epithelial Cells

To determine if sensitivity of TNBC cells to AgNPs differed based upon intrinsic subtype, a panel of BLBC (HCC70, BT-20, MDA-MB-468) and CLBC (BT-549, MDA-MB-231, MDA-MB-436, SUM159) cell lines were treated with increasing doses of AgNPs. Viability was assessed by MTT assay. For comparison, the relative sensitivities to AgNPs of non- malignant breast epithelial cells (MCF-10A, iMEC) were also assessed (Figure 1A) and IC_50_ values were calculated (Figure 1B). IC_50_ values of CLBCs ranged from 0.7–18 µg/mL, BLBCs ranged from 48–98 µg/mL, and non-cancer cell lines ranged from 110–346 µg/mL. All CLBC cell lines were significantly more sensitive than BLBC cell lines, and both were significantly more sensitive to AgNPs than non-malignant breast cells.

Next, we quantified the effect of AgNPs on clonogenic growth (Figure 1C). Cloning efficiency of CLBC cells (MDA-MB-231, BT-549, and SUM159) was significantly inhibited by AgNP at doses that had little effect on BLBC (MDA-MB-468) or non-malignant breast cells (MCF-10A and iMEC). BT-20 or HCC70 cells did not form colonies under the conditions tested and were excluded from analysis. All CLBC cell lines failed to form clones when treated with a dose of 12.5 µg/mL AgNP or higher. In contrast, cloning efficiencies of the BLBC and non-malignant cells exposed to 12.5 µg/mL AgNP were 54%, 83%, and 98%, respectively.

As a third metric to assess the cytotoxicity of AgNPs on CLBC and BLBC, proliferation of AgNP treated cells was monitored by time-lapse microscopic imaging (Figure 1D). CLBC cells treated with AgNPs doses of 7.8 µg/mL (MDA-MB-231) or 15.6 µg/mL (BT-549) or greater proliferated significantly less than vehicle-treated cells. In contrast, both BLBC cells (BT-20, MDA-MB-468) and non-malignant mammary epithelial cells (iMEC, MCF-10A) were less sensitive to AgNPs, and only the 125 µg/mL dose, the highest dose tested, had an effect on proliferation on any of these cell lines. 

All three metrics (MTT assay, clonogenic growth, proliferation) indicated that CLBC cell lines were more sensitive to AgNPs than BLBC or non-malignant mammary epithelial cells. Increased sensitivity to AgNPs aligned with a claudin-low gene signature, including genes originally defined by Prat et al. [2] and expanded by Fougner et al. [35] (Figure 2A). The expression level of a subset of mRNAs for genes associated with CLBC, including low CDH1, CLDN7, ERBB2, ESR1, KRT5, OCLN, and high VIM, ZEB1, and ZEB2 each individually correlated with the IC_50_ for AgNP treatment (determined by MTT assay) (Figure 2B). These data strongly indicate that CLBC cells are vulnerable to treatment using AgNPs.

### 3.2. Sensitivity to AgNP Exposure Does Not Correlate with Mass of AgNPs Taken Up, Inherent Sensitivity to Silver Cations, or Proliferation Rate

Having observed that CLBC cell lines were more sensitive to AgNPs than BLBC and non-malignant breast epithelial cells, we asked if increased uptake of AgNPs, inherent sensitivity to silver cations (Ag^+^), or increased proliferation rate accounted for this difference. To determine if uptake of AgNPs correlated with sensitivity to AgNPs, cells were exposed to AgNPs, and internalized silver content was analyzed by ICP-MS. The mass of AgNPs taken up by cells was heterogeneous across cell lines and significant differences were detected between cell lines (Figure 3A). MCF-10A cells took up significantly more silver than iMEC, BT-20, and MDA-MB-231 cells, while BT-549 cells took up significantly more silver than BT-20 cells. However, there was no correlation between the mass of AgNPs taken up and the AgNP IC_50_ across these cell lines (Figure 3B).

Next, we assessed the effect of AgNO_3_, used as a source of Ag^+^, on cell viability. Analysis of the IC_50_ of Ag^+^ treatment for each cell line showed variability in their sensitivity to Ag^+^ (Figure 3C). iMEC cells were more sensitive to AgNO_3_ compared to MDA-MB-231 and MCF-10A, but no significant differences were found between BLBC and CLBC cell lines. Additionally, there was no correlation between AgNP IC_50_ and AgNO_3_ IC_50_, indicating that sensitivity to Ag^+^ was not predictive of sensitivity to AgNPs (Figure 3D). 

Lastly, we examined the relationship between proliferation rate and sensitivity to AgNPs. The proliferation rate of each cell line during log-phase growth was determined and doubling time was calculated. No correlation was found between doubling time and AgNP IC_50_ (Figure 3E). These data indicate that differences in uptake of AgNPs, sensitivity to Ag^+^, and proliferation rate are insufficient to account for the high sensitivity of CLBC to AgNPs.

### 3.3. AgNPs Induce Lipid Peroxidation and 4-HNE-Histidine Adduct Formation in CLBC but Not in BLBC

Exposure to AgNPs may cause lipid peroxidation in certain cell lines or organisms [30,31,32]. Multiple studies indicate various cancers with an EMT phenotype are vulnerable to agents that induce lipid peroxidation [14,15], but differences in the sensitivity of CLBC and BLBC cells to lipid peroxidation have not been assessed. We used Liperfluo, a fluorescent probe specific for detecting lipid peroxides [36], to quantify changes in lipid peroxidation following AgNP treatment. A significant increase in fluorescent staining indicative of lipid peroxidation was detected in AgNP-treated CLBC cells (BT-549, MDA-MB-231), but not BLBC cells (HCC70, BT-20) (Figure 4A,B and Appendix A). 4-HNE is a toxic degradation product of lipid peroxides that readily forms protein adducts both of Michael addition products and Schiff bases. Therefore, to confirm that AgNPs selectively increase lipid peroxides in CLBC cells, we quantified 4-HNE histidine adducts in AgNP-treated cells (Figure 4C). We observed a dose dependent increase in 4-HNE histidine adducts in BT-549 (CLBC) cells exposed to AgNPs. In contrast, 4-HNE histidine adducts decreased in HCC70 cells. These data show that AgNPs cause lipid peroxidation in CLBC cells, but not BLBC cells, which correlate with the overall sensitivity of these TNBC subtypes to AgNP exposure.

### 3.4. AgNPs Induce Protein Aggregation, Protein Oxidation, and Proteotoxic Stress Responses in CLBC but in Not BLBC

4-HNE adducts increase protein misfolding and aggregation [37,38]. Therefore, we asked if AgNPs induce a distinct pattern of protein aggregation that correlates with relative sensitivity of CLBC and BLBC cells to AgNPs. To determine if AgNPs induce protein aggregation, cells were treated with AgNPs for 24 h, and stained with proteostat, a dye that fluoresces after intercalation into hydrophobic pockets formed by misfolded or aggregated proteins. A significant increase in fluorescent staining indicative of protein aggregation was detected in CLBC cells (BT-549, MDA-MB-231) but not in BLBC cells (HCC70, BT-20) (Figure 5A,B). Lipid peroxides also cause oxidation of thiols in proteins leading to generation of sulfenic acids [39]. We used DCP-NEt_2_C, a fluorescent probe that is specific for imaging mitochondrial protein sulfenylation [40], to assess protein oxidation following AgNP exposure. A significant increase in fluorescent staining indicative of protein oxidation was detected in AgNP treated CLBC cells (BT-549, MDA-MB-231) but not BLBC cells (BT-20, HCC70) (Figure 5C).

Accumulation of oxidized or misfolded proteins causes proteotoxic stress and can be cytotoxic. Cells will activate stress response programs to mitigate damage. This includes the integrated stress response (ISR), indicated by phosphorylation of eIF2α [41], and mitogen-activated protein kinase (MAPK) signaling pathways, indicated by phosphorylation of c-Jun N-Terminal Kinase (JNK) [42]. However, if damage is too severe and deemed irreparable, rather than acting as a pro-survival signal, ISR signaling increases expression of CHOP (CCAAT-enhancer-binding protein homologous protein), which induces apoptosis. In CLBC cells (BT-549) treated with AgNPs, p-eIF2α, p-JNK, and CHOP increased indicating activation of the ISR, and full-length caspase-7 and caspase-9 decreased, which is indicative of apoptosis (Figure 5D). No changes in ISR signaling were detected in AgNP-treated BLBC cells (HCC70). Similarly, activation of JNK and pro-apoptotic signaling was observed in AgNP treated SUM159 (CLBC) cells but not in BT-20 (BLBC) cells (Appendix A). Consistent with a slow accumulation of misfolded proteins driving toxicity, increased ISR signaling in CLBC cells was observed 24 h after AgNP exposure, but not after 3 h (Appendix A). Overall, our data support a mechanism whereby AgNPs selectively kill CLBC cells by generation of lipid peroxides and subsequent formation of 4-HNE protein adducts and oxidation of proteins, which leads to protein aggregation, proteotoxic stress, and cell death.

## 4. Discussion

TNBCs classified as CLBC/M or BLBC/BL1/BL2 have different genetics and responses to chemotherapies [2,8]. Several oncogenic drivers of these subtypes have been identified, including p53, Rb1, Ras/MAPK, and Myc [43,44], but thus far, no targeted therapeutics for these oncogenes have been effective or safe for the treatment of any form of TNBC. We posit that, in large part, this is due to two issues: (1) specifically targeted molecular aberrations may only be present in a small fraction of patients, limiting the patient population in which new treatments can be tested; (2) inhibitors of oncogenes produce severe off-target effects in non-cancer cells and tissue. These suppositions are supported by others [45]. Overcoming these issues is challenging and requires both identifying agents that exploit highly prevalent vulnerabilities specific to cancer cells themselves, and simultaneously identifying patient populations who may benefit most from targeting of those vulnerabilities. Recent studies indicate that cancers with a mesenchymal phenotype, such as CLBC, are susceptible to agents that cause oxidation of lipids or proteotoxic stress [12,13,18]. We show that by perturbing both vulnerabilities using AgNPs, CLBC cells are killed at low doses that do not affect non-malignant breast epithelial cells. Both CLBC and BLBC cell lines are more sensitive to AgNPs than non-malignant breast epithelial cells, but the AgNP IC_50_s for CLBC cell lines were approximately 5–500 fold less than those for non-malignant breast epithelial cells. In contrast, IC_50_s for BLBC cell lines were only 1.1–4-fold less than those for non-malignant cell lines. This indicates that AgNPs may have a wider therapeutic window for treatment of CLBC as compared to BLBC.

Mechanistically, we found that the difference in sensitivity of CLBC and BLBC cells to AgNPs is driven by peroxidation of lipids, protein oxidation and aggregation, and subsequent proteotoxic stress and apoptotic signaling, which are induced in AgNP-treated CLBC cells, but not BLBC cells. Although Ag^+^ present in AgNP preparations or released in cell culture media may contribute to AgNP cytotoxicity [46,47,48,49], we found no correlation between AgNP IC_50_ and Ag^+^ IC_50_, indicating that inherent sensitivity to Ag^+^ was not a factor driving the sensitivity of CLBCs to AgNPs. We also quantified the mass of AgNPs taken up by cells. We found that uptake of AgNPs by BLBC, CLBC, and non-cancer cells was variable, but there was no correlation between AgNP IC_50_ and internalized mass of AgNPs. Lastly, there was no correlation between AgNP IC_50_ and proliferation rate. This led us to infer that other factors must be causing sensitivity to AgNPs.

Analysis of mRNA profiles indicates that AgNP IC_50_ highly correlates with multiple CLBC-associated genes; however, the specific genetic drivers of AgNP sensitivity remain to be determined. CLBCs may be more primed for lipid peroxidation by AgNPs because of a lipid-signature rich in long-chain PUFAs associated with EMT [15,16,17]. Furthermore, increased uptake of exogenous PUFAs has been linked to oncogenic MYC expression and poor outcomes in CLBC [50]. The accumulation of lipid peroxides is a hallmark of a form of programmed cell death called ferroptosis, which involves iron-catalyzed generation of free radicals that cause catastrophic oxidation of phospholipids [51,52]. Preclinical studies indicate that drugs known to cause lipid peroxidation and ferroptosis are promising for treating TNBC [15,53,54]. Repurposing of clinically approved drugs, including the arthritis drug sulfasalazine [55,56] and anti-malarial artesunate [57], has shown pre-clinical efficacy by inducing lipid peroxidation in TNBC cells [58,59], but stratification into CLBC and BLBC was not performed in these studies and clinical efficacy remains to be determined. While it is unclear if currently approved drugs that increase lipid peroxidation will improve outcomes for TNBC, new ferroptosis inducers, such as imidazole-ketone-erastin show promising specificity, effectiveness, and exhibit a favorable toxicity profile, supporting use in future clinical trials [60]. It is not known if AgNPs induce ferroptosis or offer comparative advantages with regard to therapeutic index compared to small molecule ferroptosis inducers, and additional studies will be necessary to assess this. However, our data showing AgNPs specifically cause lipid peroxidation in CLBC but not BLBC indicate that greater patient stratification may lead to improved outcomes for clinical trials of other therapeutics with a similar mechanism of action.

Exposure to agents that cause accumulation of misfolded proteins in the endoplasmic reticulum, or in degradative vesicles including lysosomes and autophagosomes, can cause disruptions in protein synthesis, folding, and trafficking, leading to proteotoxicity and cell death [61]. We show that downstream of AgNP-mediated lipid peroxidation, damage propagates through CLBC cells by subsequent oxidation of proteins, generation of 4-HNE protein adducts, accumulation of protein aggregates, activation of ISR signaling, and cell death. This is consistent with studies showing that TNBC cells that have undergone EMT are more sensitive to proteotoxicity than luminal A breast cancer or non-cancer cells [13]. The basis for sensitivity to proteotoxicity in CLBC may lie in the high level of synthesis of ECM proteins including fibronectin and collagens [12,13], which places a high baseline burden on proteostasis machinery [62]. This idea is exploited clinically for multiple hematologic malignancies for which high rates of protein synthesis is a key feature of oncogenic transformation [63]. Drugs approved for clinical use for multiple myeloma and mantle cell lymphoma based upon this mechanism of action include proteasome inhibitors (bortezomib/carfilzomib) [64], E3-ubiquitin ligase inhibitors (thalidomide/lenalidomide) [65], and HDAC inhibitor (panobinostat) [66,67]. These drugs have demonstrated promising efficacy in killing mesenchymal and metastatic TNBC in vitro. Additionally, NVP-BEZ235, a dual-phosphoinositide 3-kinase (PI3K)-mammalian target of rapamycin (mTOR) inhibitor was shown by Lehmann et al. to be cytotoxic in vitro to mesenchymal TNBCs at doses that did not affect epithelial TNBCs [8]. However, clinical trials of this drug in other cancers were terminated after dose escalation studies due to severe toxicity [68]. Our previous studies established a safe, intravenous dosing window for effective treatment of MDA-MB-231 tumors (CLBC) with minimal off-target toxicity in murine models [24]. Thus, there is substantial room for development of new agents such as AgNPs that induce proteotoxic stress as a strategy to treat CLBC patients.

Previous studies support the existence of a common mechanism that drives the sensitivity of various cancers with an EMT phenotype to engineered nanomaterials. Exposure to gold nanoparticles (AuNPs) reduced expression of mesenchymal markers in ovarian cancer in vitro, leading to inhibition of tumor growth in vivo [20]. Silver nanoparticles blocked radiotherapy induced expression of EMT markers in non-small cell lung cancer (NCSLC) [22]. Titanium dioxide nanoparticles (TiO2-NPs) suppressed EMT-mediated cell migration and invasion without exhibiting cytotoxicity [69]. Gadolinium-metallofullerenol nanoparticles blocked epithelial-to-mesenchymal transition in triple-negative breast cancer (TNBC) and eliminated breast cancer stem cells (CSCs) [21]. Furthermore, we previously observed that AgNPs selectively eliminate certain TNBCs, ovarian cancer cells, and NCSLC with a mesenchymal phenotype [23,27,29,33]. Reports indicate that certain nanoparticles, including iron-core nanoparticles [70,71], titanium dioxide (TiO2) [72], ultrasmall poly (ethylene glycol)-coated silica nanoparticles [73], cobalt NPs [74], and manganese silicate nanobubbles [75], induce lipid oxidation in a variety of cell types. The cause of the general vulnerability of various cancers with an EMT phenotype to engineered nanomaterials has not been firmly established, but our data indicate that nanoparticle-induced lipid peroxidation and proteotoxicity may play a key role in this. 

Classification of breast tumors as CLBC is not part of current clinical practice, and thus transcriptomic profiling of patient tumors will need to become a routine part of breast cancer practice to select patients for AgNP treatment. However, similarities in the gene expression patterns of CLBC and highly aggressive, metaplastic breast cancers [4] suggest that the mesenchymal phenotype shared by CLBC may be associated with metaplastic characteristics, including increased invasion, metastasis, and treatment resistance [76]. It remains to be seen if AgNPs are effective at treating metaplastic breast tumors, but the SUM159 cell line studied here exhibits features of metaplastic cancers [77] and is sensitive to AgNP treatment, which supports the possibility that AgNPs may be effective for treatment of metaplastic breast cancer. 

## 5. Conclusions

Our study demonstrates previously unknown differences in the responses of mesenchymal (CLBC) and epithelial (BLBC) subtypes of TNBC to AgNPs. Specifically, we observed that AgNPs selectively induced lipid peroxidation, which drove protein oxidation and aggregation, and caused irremediable proteotoxic stress in CLBC cells without causing similar damage to BLBC cells. Furthermore, doses of AgNPs that were lethal to CLBC cells had no effect on the growth or viability of non-malignant breast epithelial cells. This study lays the foundation for the development of AgNPs as a form of precision medicine for CLBC and will guide future studies pertaining to the mechanism of action of AgNPs for cancer therapy. Additionally, our finding highlights vulnerabilities in CLBC that can be targeted by other agents, indicating that increased stratification of TNBC subtypes at the pre-clinical and clinical level may lead to improved outcomes for use of other therapeutics with a similar mechanism of action. 

## Figures and Tables

**Figure 1 cancers-13-04217-f001:**
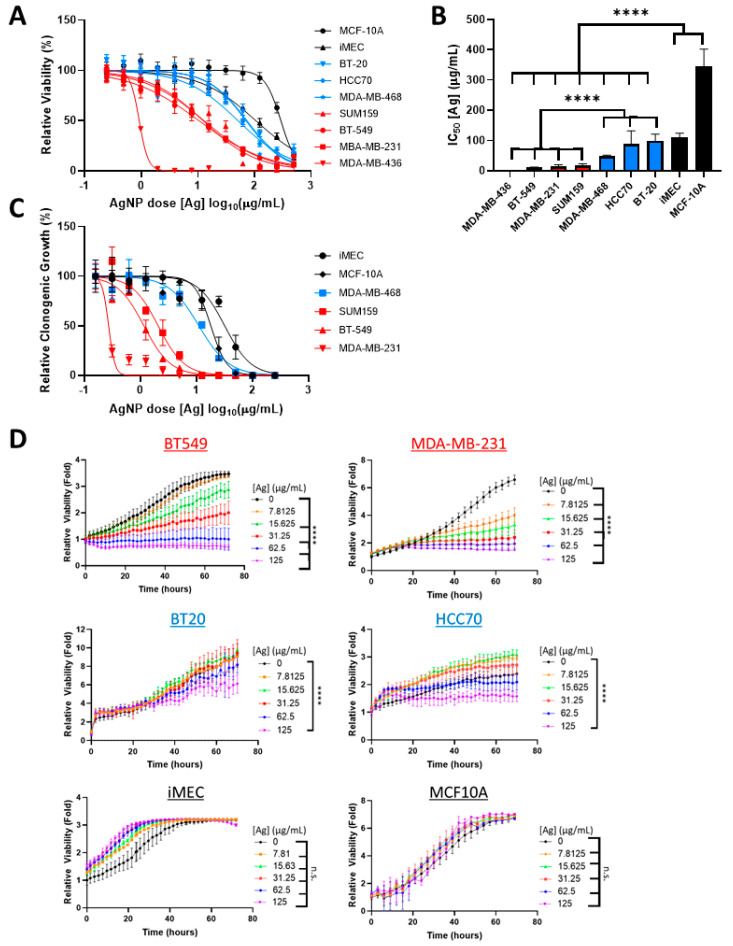
Claudin-low breast cancer (CLBC) cells are sensitive to low doses of silver nanoparticles (AgNPs), which do not affect basal-like breast cancer (BLBC) cells. (**A**) CLBC (red), BLBC (blue), and non-malignant (black) cells were treated with AgNPs for 72 h and viability was assessed by MTT assay. (**B**) IC_50_ was calculated. Data were obtained from at least four biological replicates per dose and three independent experiments per cell line. Statistical analysis was performed by two-way ANOVA followed by post hoc Tukey test. Significant differences are indicated (**** *p* < 0.0001). (**C**) Clonogenic survival of CLBC (red), BLBC (blue), and non-malignant (black) cells was assessed after AgNP treatment. Data were obtained from three biological replicates per dose and are representative of duplicate independent experiments per cell line. (**D**) Proliferation was assessed by Incucyte time-lapse imaging of the indicated cell lines over 72 h with or without AgNPs. Data were obtained from at least four biological replicates per dose and are representative of three independent experiments per cell line. Statistical analysis was performed by two-way ANOVA followed by post hoc Tukey test. Significant differences are indicated (**** *p* < 0.0001).

**Figure 2 cancers-13-04217-f002:**
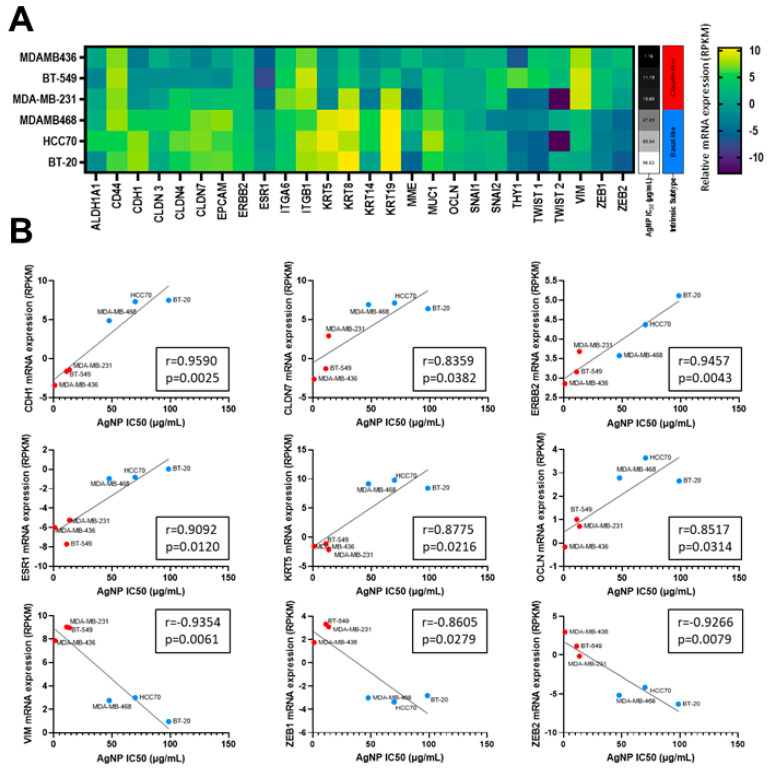
Sensitivity to AgNPs aligns with a CLBC gene signature. (**A**) mRNA expression profiles from the Broad-Novartis Cancer Cell Line Encyclopedia (CCLE) show the expected clustering of CLBC and BLBC cell lines. (**B**) AgNP IC_50_ was plotted versus mRNA expression of each gene from (**A**). Pearson’s correlation coefficient (r) was calculated as shown. mRNA expression vs. IC_50_ is shown for the subset of genes in (**A**) with expression that significantly correlated (*p* < 0.05) with sensitivity to AgNPs.

**Figure 3 cancers-13-04217-f003:**
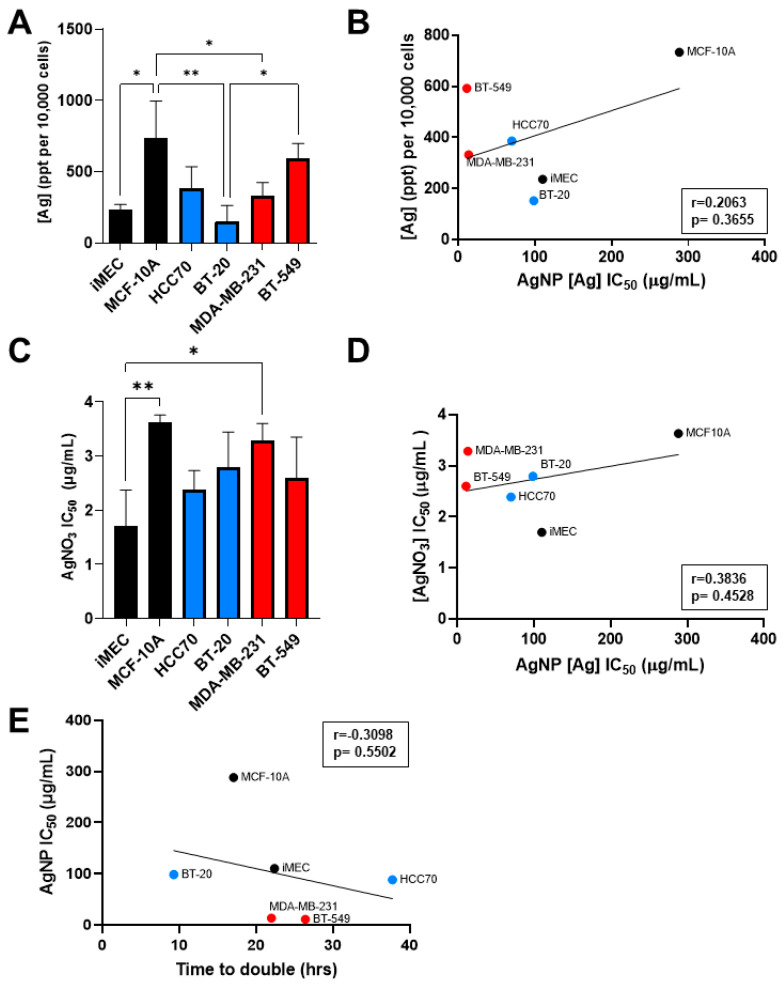
Sensitivity to AgNPs does not correlate with uptake of silver, sensitivity to silver cations, or proliferation rate. (**A**) Uptake of AgNPs was quantified by ICP-MS from three biological replicates per cell line and (**B**) plotted versus AgNP IC_50_. (**C**) Cells were exposed to Ag^+^ (in the form of AgNO_3_) for 72 h, viability was assessed by MTT assay, and IC_50_ was calculated. Data were obtained from six biological replicates per dose and three independent experiments per cell line. (**D**) AgNO_3_ IC_50_ was plotted versus AgNP IC_50_. (**E**) Proliferation rate was determined from Figure 2B and plotted versus AgNP IC_50_. (**A**,**C**) Statistical analysis was performed by one-way ANOVA followed by post hoc Tukey test. Significant differences are indicated (* *p* < 0.05; ** *p* < 0.01). (**B**,**D**,**E**) Pearson’s correlation coefficient (r) was calculated as shown. No significant correlation was detected in any data set (*p* > 0.05).

**Figure 4 cancers-13-04217-f004:**
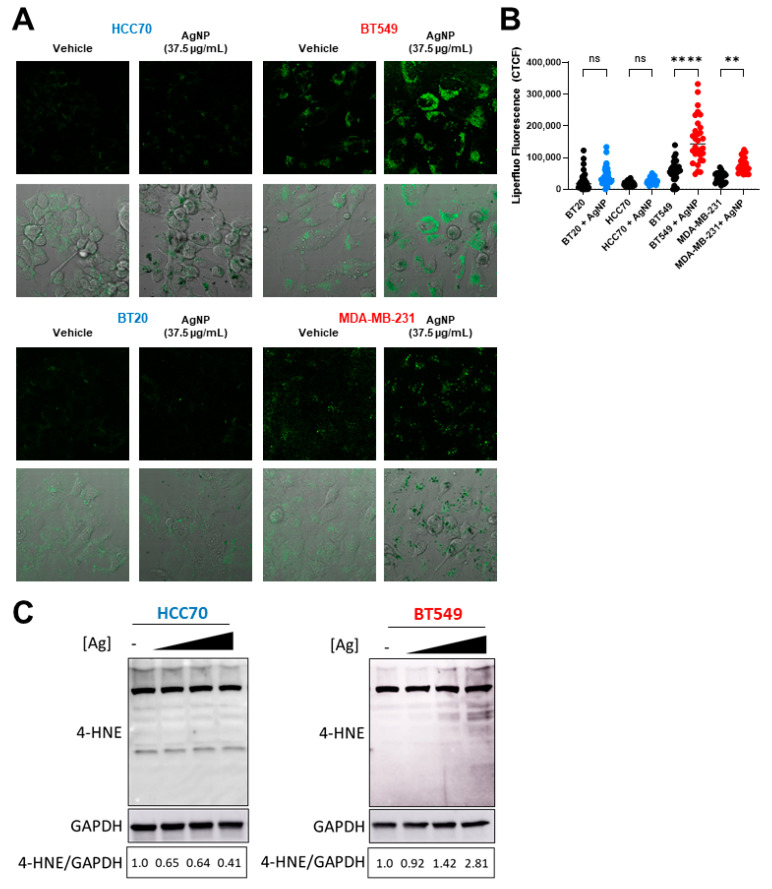
AgNPs induce lipid peroxidation and 4-HNE adduct formation in CLBC but not BLBC. To assess lipid peroxidation, BLBC (blue) and CLBC (red) cells were treated with AgNPs for 24 h, stained with Liperfluo, and fluorescence assessed by confocal microscopy (**A**,**B**). (**A**) Microscopy results shown are representative images from three independent experiments with at least three random fields assessed per sample. (**B**) Fluorescence intensity was quantified in at least 10 cells per field and statistical analysis was performed by one-way ANOVA followed by post hoc Tukey test. Significant differences are indicated (** *p* < 0.01; **** *p* < 0.0001). (**C**) 4-Hydroxynonenal (4-HNE) was assessed by Western blot in cells treated with (left to right) 0, 18.75, 37.5, or 75 µg/mL AgNPs for 24 h. Densitometry was performed and the ratio of 4-HNE to GAPDH was calculated as indicated. The results are representative duplicate independent experiments.

**Figure 5 cancers-13-04217-f005:**
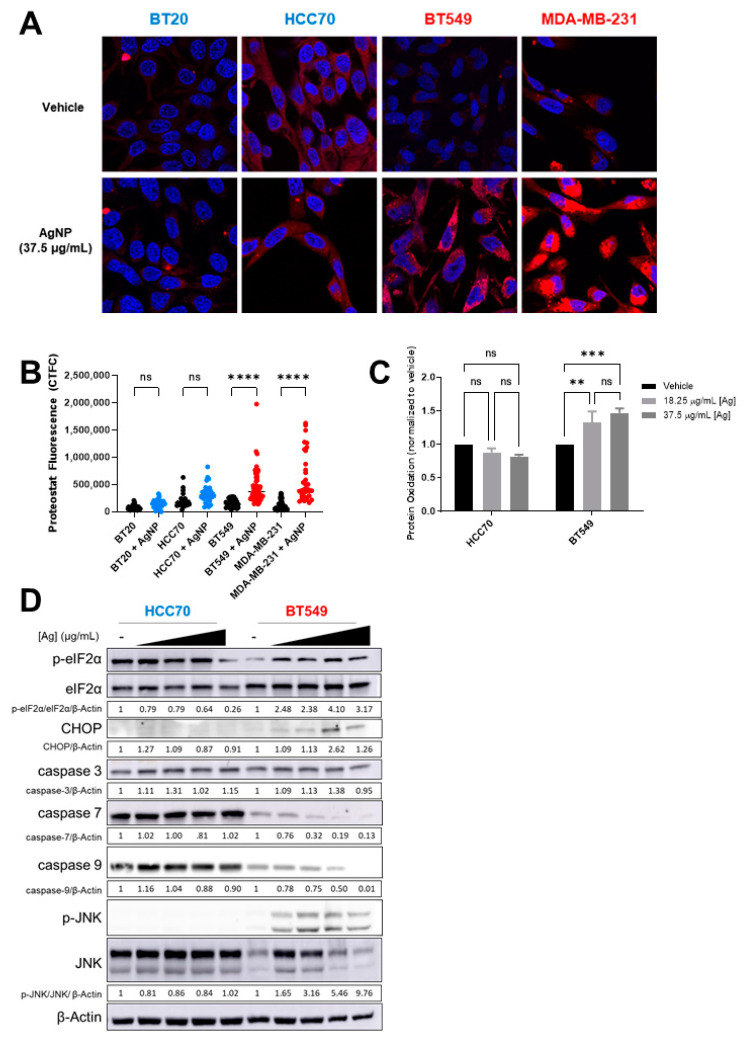
AgNPs induce protein aggregation and protein oxidation and activate proteotoxic stress responses in CLBC but not BLBC. (**A**,**B**) To assess protein aggregation, BLBC (blue) or CLBC cells (red) were treated with AgNPs for 24 h, stained with Proteostat aggresome dye, and fluorescence was measured using confocal microscopy. (**A**) Microscopy results shown are representative images from three independent experiments with at least three random fields assessed per sample. (**B**) Fluorescence intensity was quantified in at least 10 cells per field and statistical analysis was performed by one-way ANOVA followed by post hoc Tukey test. Significant differences are indicated (**** *p* < 0.0001). (**C**) To assess protein oxidation, cells were treated with AgNPs for 24 h, stained with DCP-Net2C, and fluorescence assessed by flow cytometry. Data were obtained from three independent experiments. Statistical analysis was performed by two-way ANOVA followed by post hoc Tukey test. Significant differences are indicated (** *p* < 0.01; *** *p* < 0.001). (**D**) To assess ISR signaling, cells were treated with 0, 18.25, 37.5, or 75 µg/mL AgNPs for 24 h and protein expression was evaluated by Western blot. Densitometry was performed and the ratio of peIF2α/eIF2α or each individual protein to β-actin was calculated as indicated. The results are representative duplicate independent experiments.

**Table 1 cancers-13-04217-t001:** Media formulations.

Cell Line	Media Formulation
BT-20	EMEM supplemented with penicillin (250 units/mL), streptomycin (250 μg/mL), 2 mM L-glutamine, and 10% fetal bovine serum
BT-549	RPMI supplemented with penicillin (250 units/mL), streptomycin (250 μg/mL), and 10% fetal bovine serum
HCC70	RPMI supplemented with penicillin (250 units/mL), streptomycin (250 μg/mL), and 10% fetal bovine serum
iMEC	DMEM/F12 supplemented with 10 µg/mL insulin, 20 ng/mL human epidermal growth factor (hEGF), and 0.5 μg/mL hydrocortisone, and 10% fetal bovine serum
MCF-10A	DMEM/F12 supplemented with penicillin (250 units/mL), streptomycin (250 μg/mL), 2 mM L-glutamine, 10 μg/mL insulin, 20 ng/mL EGF, 0.5 μg/mL hydrocortisone, and 100 ng/mL cholera toxin, and 5% heat-inactivated horse serum
MDA- MB-231	DMEM/F12 supplemented with penicillin (250 units/mL), streptomycin (250 μg/mL), 2 mM L-glutamine, and 10% fetal bovine serum
MDA- MB-436	DMEM supplemented with penicillin (250 units/mL), streptomycin (250 μg/mL), 2 mM L-glutamine, and 10% fetal bovine serum
MDA- MB-468	DMEM supplemented with penicillin (250 units/mL), streptomycin (250 μg/mL), 2 mM L-glutamine, and 10% fetal bovine serum
SUM-159	HAM’s F12 supplemented with penicillin (250 units/mL), streptomycin (250 μg/mL), 2mM L-glutamine, 5 μg/mL insulin, 1 μg/mL hydrocortisone, 10 μM HEPES, and 5% fetal bovine serum

## Data Availability

Data sets are available as detailed in the Methods section.

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
