# Peer review of "Low Doses of Silver Nanoparticles Selectively Induce Lipid Peroxidation and Proteotoxic Stress in Mesenchymal Subtypes of Triple-Negative Breast Cancer"

_cancers, 2021, doi:10.3390/cancers13164217_

Round 1
Reviewer 1 Report
The authors have done excellent research on the effect of silver nanoparticles on triple negative breast cancer cell lines, focusing on the differences between the mesenchymal and epithelial subtypes.
The research carried out has been excellent with a total of 9 cell lines and a few techniques that consolidate the results and conclusions presented by the authors.
Some comments towards the authors are:
1.- Why is nothing mentioned about ERbeta? Not all cell lines that have been used in research are estrogen receptor beta negative. Is there a relationship between the results obtained and this estrogen receptor subtype?
2.- In line 235 of page 6 there is a formula with a different format than the rest of the manuscript, it would be better to modify it.
Author Response
Reviewer 1: The authors have done excellent research on the effect of silver nanoparticles on triple negative breast cancer cell lines, focusing on the differences between the mesenchymal and epithelial subtypes. The research carried out has been excellent with a total of 9 cell lines and a few techniques that consolidate the results and conclusions presented by the authors.
Response: We greatly appreciate these kind words and thank the reviewer for taking the time to provide feedback on our manuscript.
Reviewer 1: Why is nothing mentioned about ERbeta? Not all cell lines that have been used in research are estrogen receptor beta negative. Is there a relationship between the results obtained and this estrogen receptor subtype?
Response: The role of ER-beta in TNBC remains debatable due to poor quality of many antibodies used for immunohistochemistry or western blotting. Overall, studies using validated antibodies or mRNA expression to quantify ER-beta levels find that expression of this receptor is heterogenous, with various studies concluding 0-30% of TNBCs express ER-beta. It is generally believed that ER-beta expression is associated with a better prognosis for TNBC patients. However, expression of ER-beta is not known to be associated with any specific subtype of TNBC and therefore the role of this receptor was not assessed in the outcomes we observed with regard to treatment of CLBC and BLBC with AgNPs.
Reviewer 1: In line 235 of page 6 there is a formula with a different format than the rest of the manuscript, it would be better to modify it.
Response: The journal’s editorial staff performed this formatting, and we will accede to the journal’s editorial policies with regard to final formatting.
Reviewer 2 Report
In the manuscript by Snyder et al. the authors show AgNPs are a specific treatment for claudin-low breast cancer (CLBC), which provides a novel potential treatment for triple-negative breast cancer, especially CLBC. I would suggest only minor changes.
Page 10, line 331-333:" Because exposure to AgNPs may cause lipid peroxidation (30-32), we examined if AgNPs selectively induced lipid
peroxidation in CLBCs." Please change to " We examined if AgNPs selectively induced lipid peroxidation in CLBCs, because exposure to AgNPs may cause lipid peroxidation (30-32)."
Page 10, line 338-339:" both Michael addition products and 338
Schiff bases." Please change to " both of Michael addition products and 338 Schiff bases."
Page 13, line 430:"Lastly" Please change to "Additionally"
Some citations are missing
Page 13, line 405-407:"In large part, this is due to two issues: (1) specifically targeted molecular aberrations may only be present in a small fraction of patients, limiting the patient population in which new treatments can be tested; (2) inhibitors of oncogenes produce severe off-target effects in non-cancer cells and tissue." are shown without citation or explanation.
Additionally: I greatly enjoyed reading the manuscript. It will be a valuable resource for further research on Nanoparticles as novel therapeutics for breast cancer. I hopr the authors’ understanding if my comments above focus on what I think can/should be improved, without stating often enough how positive I felt about the paper in general. Don’t misinterpret this as a negative attitude - the overall impression is very strongly positive. Hence I express my strong support for publication.
Author Response
Reviewer 2: Page 10, line 331-333:" Because exposure to AgNPs may cause lipid peroxidation (30-32), we examined if AgNPs selectively induced lipid peroxidation in CLBCs." Please change to " We examined if AgNPs selectively induced lipid peroxidation in CLBCs, because exposure to AgNPs may cause lipid peroxidation (30-32)."
Response: We agree this phrasing is cumbersome and have restructured the paragraph to reduce wordiness.
Reviewer 2: Page 10, line 338-339:" both Michael addition products and 338 Schiff bases." Please change to " both of Michael addition products and 338 Schiff bases."
Response: The change has been made.
Reviewer 2: Page 13, line 430:"Lastly" Please change to "Additionally"
Response: The change has been made.
Reviewer 2: Some citations are missing. Page 13, line 405-407:"In large part, this is due to two issues: (1) specifically targeted molecular aberrations may only be present in a small fraction of patients, limiting the patient population in which new treatments can be tested; (2) inhibitors of oncogenes produce severe off-target effects in non-cancer cells and tissue." are shown without citation or explanation.
Response: These suppositions are our own based upon our expertise in TNBC. We have restructured the phrasing to emphasize that this is or own reasoning, and will cite a recent review article on the topic to support our claims [Yin L, Duan JJ, Bian XW, Yu SC: Triple-negative breast cancer molecular subtyping and treatment progress. Breast Cancer Res 2020, 22(1):61].
Reviewer 2: I greatly enjoyed reading the manuscript. It will be a valuable resource for further research on Nanoparticles as novel therapeutics for breast cancer. I hopr the authors’ understanding if my comments above focus on what I think can/should be improved, without stating often enough how positive I felt about the paper in general. Don’t misinterpret this as a negative attitude - the overall impression is very strongly positive. Hence I express my strong support for publication.
Response: We thank the reviewer for these kind words and greatly appreciate the efforts made to improve the quality of our manuscript.